# Multiple Laparoscopic Liver Resection for Colorectal Liver Metastases

**DOI:** 10.3390/cancers15020435

**Published:** 2023-01-10

**Authors:** Alexandra Nassar, Stylianos Tzedakis, Alix Dhote, Marie Strigalev, Romain Coriat, Mehdi Karoui, Anthony Dohan, Martin Gaillard, Ugo Marchese, David Fuks

**Affiliations:** 1Department of Hepato-Pancreatic-Biliary and Endocrine Surgery, Cochin Hospital, Assistance Publique-Hôpitaux de Paris Centre, Université Paris Cité, 75014 Paris, France; 2Department of Gastroenterology and Digestive Oncology, Cochin Hospital, Assistance Publique-Hôpitaux de Paris Centre, Université Paris Cité, 75014 Paris, France; 3Department of General Digestive Surgery and Cancerology, Hopital Européen Georges Pompidou, Université Paris Cité, 75015 Paris, France; 4Department of Radiology, Cochin Hospital, Assistance Publique-Hôpitaux de Paris Centre, Université Paris Cité, 75014 Paris, France

**Keywords:** laparoscopic liver resections, colorectal liver metastases, multiple laparoscopic resections

## Abstract

**Simple Summary:**

Colorectal liver metastases are multiple in 80% of cases, and surgical resection is still the best treatment known in terms of survival. Laparoscopic multiple liver resections are yet not recommended, and no dedicated comparative study has been published. This literature review aimed to assess feasibility of laparoscopic liver resection for multiple colorectal liver metastases, whether by parenchymal-sparing multiple resections or two-stage resections. The purpose of this review is to guide the implementation of this minimal invasive technique for multiple colorectal liver metastases.

**Abstract:**

Over the past decades, liver cancer’s minimally invasive approach has primarily become as a new standard of oncological care. Colorectal liver metastases (CRLM) are one of the most developed indications of laparoscopic liver resection (LLR). CRLM resection is still the best treatment known in terms of survival. As multiple CRLM are found in up to 80% of cases at diagnosis (Manfredi S. and al, Annals of Surgery 2006), a lot of possible technical management approaches are described. With the development of the parenchymal-sparing strategy, multiple concomitant laparoscopic liver resections (LLR) are gaining acceptance. However, no recommendation is available regarding its indications and feasibility. Also, laparoscopic two-stage hepatectomy is developing for bilobar CRLM, and this also does not have established recommendation. The purpose of this paper was to highlight novelty and updates in the field of multiple minimally invasive liver resections. A review of the international literature was performed. The feasibility of laparoscopic concomitant multiple LLR and two-stage hepatectomy for CRLM as well as their outcomes were discussed. These clarifications could further guide the implementation of minimal resection in multiple colorectal liver metastases therapies.

## 1. Introduction

During the past decades, the laparoscopic approach in liver surgery has gained popularity, due to its various advantages compared to open surgery, which has led to its acceptance as a future new standard of care [1]. Yet, laparoscopic liver resection (LLR) has been one of the latest indications of laparoscopy, due to its prolonged learning curve and technicality [2]. 

Knowing that liver resection is the is the only treatment that currently offers a chance of long-term survival for colorectal liver metastases (CRLM) [3], the laparoscopic approach has been widely developed for CRLM, and LLR has shown its benefits compared to the open approach in this indication [4,5,6]. Paradoxically, despite two decades of the diffusion of laparoscopy, many surgeons do not perform more than 4–5 concomitant LLR for CRLM, and experts are still reluctant to recommend it [7,8]. This statement is due to the theoretical difficulty of performing multiple LLR and the lack of evidence in the literature, as many studies have a median number of one or two lesions resected at once [9].

However, the one-stage parenchymal-sparing strategy has shown its benefits compared to major hepatectomy [10,11] and to two-stage hepatectomy [12] in the open approach. Parenchymal-sparing hepatectomy in the laparoscopic approach has also shown its benefits [13,14,15]. Moreover, the better understanding of liver anatomy with the development of intraoperative ultrasonography in laparoscopy [16,17,18] should allow multiple concomitant LLR in CRLM. 

The purpose of this article was to highlight updates in the field of multiple concomitant LLR for colorectal liver metastases in order to clarify its indications and help further its implementation. The international literature was reviewed and compared to our team’s experience to discuss several topics: feasibility of multiple concomitant LLR for CRLM; available guidance techniques in laparoscopy for liver surgery; Short-term and long-term outcomes after multiple LLR for CRLM; feasibility and results of laparoscopic two-stage hepatectomy.

## 2. Feasibility of Multiple Concomitant LLR

In the study from Russolillo et al. [19], the authors aimed to assess the best outcomes in LLR based on surgical difficulty by using benchmarking. In this trial, 819 patients who underwent multiple LLR (i.e. more than one resection during the same intervention) were described, including 438 who experienced CRLM. Unfortunately, no details about the exact number of resections performed was mentioned. In the meta-analysis conducted by J Kalil et al. [14], which aimed to assess feasibility and the limitation of laparoscopic parenchymal-sparing hepatectomy, included 10 studies, from which 92 patients reported having undergone multiple LLR. In this trial, the highest number of resections reported was seven. Those two studies did not only include CRLM, which represented 58% of patients in the J Kalil et al. study. Other indications were hepatocellular carcinoma, metastatic neuroendocrine tumors, liver adenoma, and intrahepatic cholangiocarcinoma. 

Only three studies specifically compared multiple and single laparoscopic resections for CRLM [20,21,22], and one study specifically described multiple LLR for CRLM [23]. Overall, 271 patients underwent multiple concomitant LLR for CRLM in these studies, including 69 (25.5%) patients with three to five, and 24 (8.9%) patients with more than five resections. Lesions were bilobar in 153 (56.5%) patients [20,21,23]. No difference was found regarding the maximal tumor size between multiple and single LLR [21,22]. 

The type of hepatectomy [20,21,22] is described in Table 1. Most of the patients underwent multiple atypical LLR (126 patients). For patients who underwent resections in the right lobe of the liver, 3 patients underwent multiple anterior LLR, and 11 underwent posterior LLR. The majority of the studies [20,22,23] included only limited multiple resections. No difference was found in terms of grade of difficulty according to the IMM classification [24] according to the number of LLR in Nassar et al. [22]. However, multiple LLR were significantly less associated with grade III than with single resections in the Russolillo et al. trial [19].

Compared to single resections, multiple LLR did not impact blood loss [20,22], except in the series reported by D’Hondt et al. [21]. Significantly higher blood loss for bilobar CRLM (250 mL vs. 100 mL) was described in this study. However, this result was also associated with a significantly higher rate of major resections in the bilobar LLR group (32 out of 36 patients), while other studies only reported minor hepatectomies in both multiple and single resection groups. 

The conversion rate was not influenced by the number of LLR [20,21,22], with only 11 (4.1%) patients who required a conversion to open. As expected, operative time was significantly longer for multiple LLR than for single LLR in all studies. Operative time also significantly increased with the number of LLR, as described in Nassar et al [22] (175.3 min vs. 200.4 min vs. 234.1 min for <3, 3–5, >5 resections respectively, *p* = 0.039). Table 2 summarizes the per-operative outcomes for multiple LLR.

In our experience, 20% of patients considered for multiple LLR needed more than three concomitant resections. Patients who underwent multiple LLR had a mean number of 2.8 lesions. In 50% of cases, multiple LLR involved only multiple atypical resections. Conversion occurred in less than 10% of patients. Operative time steeply increased with the number of LLR performed.

Overall, multiple concomitant LLR for CRLM seem to be feasible and safe. The tumor location, maximal size, or number should not be an indication to perform a single larger resection in place of multiple small resections. Parenchymal-sparing strategy seems feasible by laparoscopy for multiple CRLM. However, operative time seems to increase with the number of resections performed, but without impacting the conversion rate or blood loss.

## 3. Ultrasonography and Other Operative Guidance in Multiple LLR for CRLM

Intraoperative ultrasound (IOUS) is nowadays well established in liver surgery to help identify anatomic landmarks and guide liver resection. Laparoscopic IOUS is particularly interesting in LLR and used to compensate for the lack of palpation [18]. Figure 1. shows an example of laparoscopic IOUS performed by our team for multiple colorectal liver metastases localization and their relation to anatomical structure. Due to its technicality and positioning difficulties, laparoscopic IOUS can appear to be more challenging than in an open approach, especially in multiple CRLM cases. However, its performance for staging liver tumors in laparoscopy is similar compared to an open approach [25], and it should be integrated into the surgeons’ habits and formation [26]. Some expert teams, in particular in Italy, proposed specific masterclasses on IOUS during liver surgery.

IOUS has demonstrated its accuracy in bilobar multiple one-stage resections for CRLM in the open approach [17]. No study has specifically been conducted to determine the accuracy of IOUS in multiple LLR for CRLM. However, IOUS was always described in different studies investigating multiple LLR [20,21,22] to guide parenchymal transection. Also, the IOUS map technique has been described recently in laparoscopy [27] and was performed in 25 patients who underwent multiple concomitant LLR. This study confirms the technical feasibility of IOUS for multiple LLR, and also suggests its effectiveness to prevent bleeding, with no major bleeding (>1000 mL) reported. 

Indocyanine green (ICG) has also gained popularity as an intraoperative aid to delineating segmental boundaries and CRLM locations in LLR [28]. Showing margins of the tumors, ICG also improve complete R0 resection. Unfortunately, no study has been conducted to investigate its efficiency for multiple LLR for CRLM, and most studies published do not include multiple resections. Lu et al. [29] described ICG navigation in LLR, which included eight patients with bilobar involvement, and showed its efficiency in this indication. Figure 2 shows an example of the use of ICG for bilobar CRLM in a laparoscopic approach performed by our team.

However, ICG has some limitations, mostly due to its inefficiency to identify deep lesions in the liver parenchyma [30]. However, some green staining during the parenchymal transection can guide the surgeon intraoperatively. If IOUS still offers the best assistance to locate deep lesions [30], ICG has demonstrated its capacity to have a real-time visualization of tumor margins when they are peripherical. Both IOUS and ICG are capable of detecting lesions that are not visible on preoperative CT-scans [30]. As a consequence, both IOUS and ICG could modified the operative strategy and improve complete resection. 

Intraoperative real-time navigation by 3D models based on preoperative CT-scan models has demonstrated its help in an open approach for parenchymal-sparing hepatectomy in multiple CRLM [31] by estimating future liver remnants. Three-dimensional modelling has not yet been validated for the laparoscopic approach, but it could help to plan complex LLR, such as multiple resections [32]. However, 3D models could be unreliable if lesions are not detected by the preoperative CT-scan, and they should always be combined with IOUS.

The description of advantages and disadvantages of these guidance techniques are presented in Table 3.

To sum up, IOUS should remain as an aid to determine a tumor’s location and burdens, but further studies should be conducted to validate indocyanine green and/or 3D modeling for multiple LLR.

## 4. Postoperative Short-Term Outcomes after Multiple LLR

No dedicated study has been made to compare the laparoscopic and open approach for multiple CRLM. However, a OSLO-COMET randomized trial [4], which compared open and laparoscopic parenchyma-sparing resections on short-term outcomes for CRLM, included multiple resections and did not reveal it as a factor of worse outcomes in the laparoscopic group.

When compared to single resections, multiple LLR were not associated with higher postoperative overall morbidity [20,21,22] or 90-day mortality [21,22]. Multiple LLR were even associated with significantly less liver failure in Kazaryan et al. [20] (0% vs. 8.3% for single resections), but the single LLR group in this study included 27.8% of major hepatectomy outcomes and only one in the multiple LLR group. The number of LLR did not impact the length of hospital stay [22]. The number of LLR also did not influence the achievement of the textbook outcomes [22,33]. Table 4 summarizes the postoperative outcomes for multiple LLR. 

However, Russolillo et al. [19] described multiple concomitant LLR as an independent risk factor for morbidity in multivariate analysis, along with difficulty in the IMM grade of resection, bowel resection and cirrhosis, whereas it was not associated with major morbidity (i.e. Clavien III–V) [34]. Also, in this trial, patients who underwent multiple LLR had significantly higher risks of pulmonary infection (3.1%) and bowel complications (2.6%) compared to those who had single LLR. However, 11.3% of patients who had multiple LLR also had bowel resection, and the proportion of minor or major resection in the multiple LLR subgroup was unknown.

All in all, results concerning postoperative pulmonary complications are conflicting. A higher risk of pulmonary infection could be due to longer operative time [35]. However, multiple LLR for CRLM does not seems to increase major complications or postoperative mortality. 

## 5. Oncological Outcomes after Multiple LLR for CRLM

As CRLM are numerous at the diagnosis in 80% of patients [36], and liver resection is still the best-known treatment for survival [3], many studies have investigated the impacts of multiple liver resection on long-term outcomes for CRLM. 

Montalti et al. [15] described that multiple liver tumors were significantly associated with the risk of R1 margins (<1 mm), but the number of LLR performed was not specifically studied. In the three papers specifically studying multiple concomitant LLR for CRLM, the rate of R0 surgical margins was not significantly different between single and multiple LLR [20,21,22]. Lu et al. [29] described ICG in multiple LLR, but failed to show the difference regarding R0 margins compared to the non-ICG group. 

No dedicated study has been made to compare the outcomes of a laparoscopic or an open approach for multiple CRLM. However, a recent randomized trial from Aghayan et al. [37] compared long-term outcomes between laparoscopic and open parenchymal-sparing resections for CRLM, and their findings suggested that multiple (>1) and bilobar CRLM resections did not impact overall (OS) or recurrence-free survival (RFS) in this cohort. 

A trial by Bolton et al. [38] described that complex metastatic disease (defined by at least four unilobar CRLM or at least two bilobar CRLM) had a 5-year survival rate of 37% after the open approach resection. A recent multicentric study [39] described long-term outcomes in 142 patients who underwent LLR for CRLM, with a 37.1% 5-year overall survival (OS) and a median survival of 39 months. Aghayan et al. [23] described a 5-year OS of 44% in 80 patients in a multiple LLR subgroup (defined by more than two concomitant resections) who had no extrahepatic metastases. 

When compared to single resections, only two studies described long-term outcomes [20,21] of multiple concomitant LLR in 140 patients with CRLM. Multiple concomitant LLR did not seem to impact OS nor RFS when compared to single resections in those studies. 

Five-year OSs and RFSs are described in Table 4.

Recurrence was a liver recurrence in 74 (52.9%) patients, and liver only occurred in 43 (30.7%) patients in these two studies. Kazaryan et al. [20] described the possibility of repeat surgical procedures to treat liver recurrence for 35% of patients, without significant difference compared to single resections (25%).

In short, multiple concomitant resections do not seem to impact R0 margins rate or oncological outcomes compared to single resections. Parenchymal-sparing multiple LLR could allow for the simplest repeat hepatectomy for recurrence [23]. 

## 6. Two-Stage Laparoscopic Hepatectomy

For patients with initially unresectable bilobar extensive CRLM (i.e. multiple lesions which cannot be resected in upfront single stage surgery with R0 margins), the realization of one-stage hepatectomy is limited, due to the risk of liver failure. For those patients, two-stage hepatectomy (TSH) improved their resectability [40] and demonstrated its advantages in an open approach [41,42]. 

Seven studies described the role of the laparoscopic approach in TSH for CRLM [43,44,45,46,47,48,49]. Descriptions of the patients included in these studies are detailed in Table 5.

Overall, 131 patients were described in those studies for a median number of lesions of 5.3 CRLM [43,44,45,47], with 119 patients who underwent laparoscopic first-stage hepatectomy, and 87 who underwent laparoscopic second-stage hepatectomy. Between the two stages, 63 patients had portal vein embolization, and 7 patients had right portal vein ligation during the first stage. The second stage was performed after a mean interval of 3.0 months [43,44,45,47]. However, 18 patients dropped out between the two stages, due to tumor progression (16 patients) or insufficient future liver remnant volume (2 patients). A total of 4 patients (3.4%) in the first stage and 11 patients (12.6%) in the second stage required conversion to an open approach. The first-stage hepatectomies consisted mostly in atypical left resections (107 patients, 89.9%), and the second-stage hepatectomies consisted mostly in right hepatectomy (71 patients, 81.6%) or right-extended hepatectomy (10 patients, 11.5%). 

Compared to open TSH [44,46], laparoscopic TSH had significantly less blood loss, significantly shorter length of hospital stay, and less overall postoperative complications for both the first and second stage when realized in laparoscopy. 

In our experience, all patients with totally laparoscopic TSH had right hepatectomy or right extended hepatectomy as the second stage. More than 90% of patients underwent PVE before the second hepatectomy. The 90-day morbidity rate after the first stage was less than 5%, with no major complications (Clavien III–IV). After the second stage, the 90-day morbidity was around 30%. Postoperative 90-day mortality was nil after both the first and second stage. 

In terms of long-term outcomes, the OS and RFS was not different from the open TSH [43,44], with a significantly higher possibility to perform repeat hepatectomy for liver recurrences [44] (58.8% of liver recurrence was treated by repeat hepatectomy in laparoscopic TSH vs. 11.8% in the open approach). Overall survival was significantly better for the patients who completed the two stages than for the patients who did not complete the second stage. 

Overall, for bilobar extended CRLM that have been initially considered unresectable, laparoscopic TSH is feasible and safe, with similar oncological outcomes to the open approach. Also, as for multiple concomitant resections, laparoscopic approach seems to simplify repeat hepatectomy for hepatic recurrence. 

## 7. Conclusions

CRLM are one of the most developed indications of LLR. Still, surgeons tend to perform multiple liver resections in the open approach. Most of the large studies comparing the laparoscopic to the open approach for CRLM do not include multiple resections. Indeed, most surgeons favor performing LLR unilobar metastases and a limited number of nodules in the context of nonrandomized trials [50]. This review suggests that multiple concomitant LLR for CRLM seems to be feasible and safe, without impacting major short-term or long-term outcomes. The feasibility of laparoscopic liver resection does not seem to be affected by the number of lesions. However, the operative time increases with the number of resections, and could impact the postoperative pulmonary complication rate. Thus, from our point of view, for patients requiring more than five concomitant resections, laparoscopy should be considered for those selected patients and managed in expert centers. In our experience, patients with multiple CRLM have been selected to undergo multiple resections or TSH in laparoscopy when they have no history of upper abdominal surgery performed in an open approach or anesthetic contraindication to laparoscopy, as well as when no difficult resection is needed (venous reconstruction / bile duct resection). Laparoscopic TSH is also a safe alternative for initially unresectable bilobar CRLM. Moreover, laparoscopic approach for multiple CRLM, resected in one or two-stages, could improve the feasibility of repeat hepatectomy for liver recurrence of CRLM. Further dedicated studies, and more prospective comparative studies, are needed to confirm those findings. Meta-analysis on this matter should also be performed when enough material is available.

## Figures and Tables

**Figure 1 cancers-15-00435-f001:**
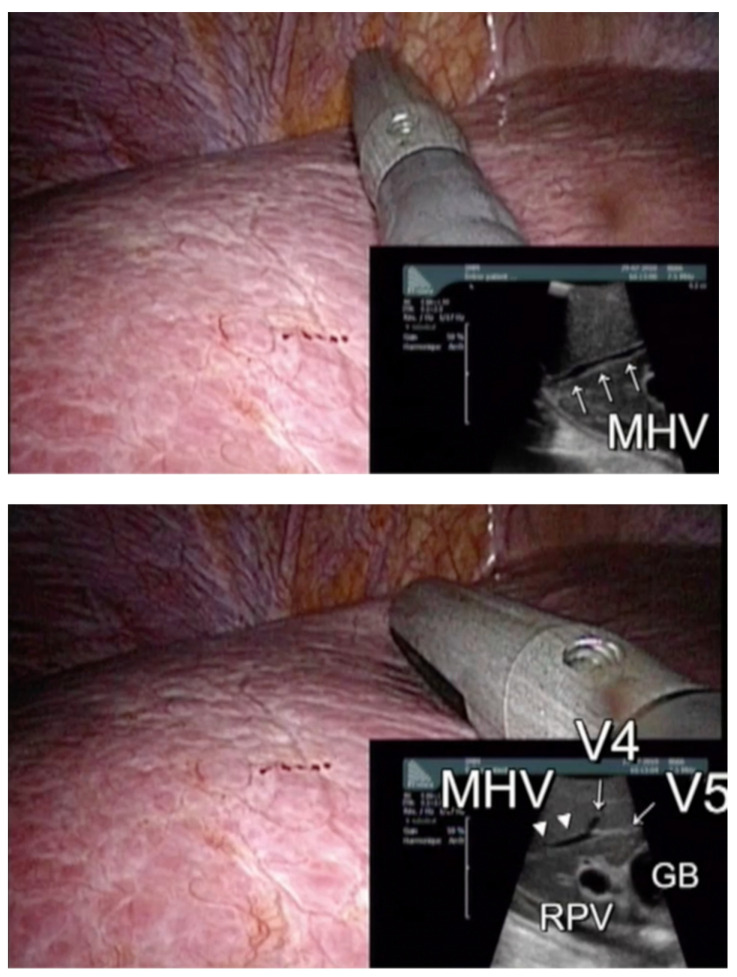
Laparoscopic intraoperative ultrasonography guidance. (MHV: median hepatic vein; RPV: right portal vein; GB: gall bladder; V4 and V5: hepatic veins of segment 4 and 5).

**Figure 2 cancers-15-00435-f002:**
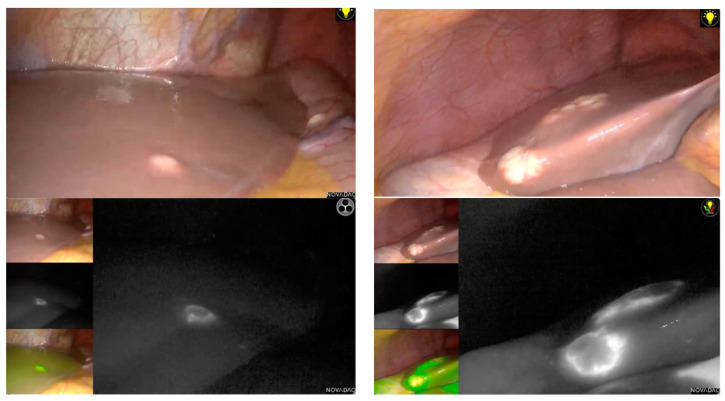
Laparoscopic use of ICG for multiple colorectal liver metastases. This figure is a case of multiple bilobar CRLM (upper images), and shows the effectiveness of laparoscopic ICG to determine CRLM location and margins, to ensure complete resection.

**Table 1 cancers-15-00435-t001:** Number and type of resections described in concomitant multiple LLR for CRLM.

Variable	Number of Patients Reported [20,21,22,23]				
Number of LLR					
2	178 (65.7%)				
3–4	69 (25.5%)				
≥5	24 (8.9%)				
**Variable**	**Number of Patients Reported**	**Kazaryan et al. [20]** **(n = 104)**	**D’Hondt et al. [21]** **(n = 36)**	**Nassar et al. [22]** **(n = 39)**	**Aghayan et al. [23]** **(n = 92)**
Type of LLR					Not described
Left lateral sectionectomy with right atypical	29	20	9		
Left hemihepatectomy with right atypical	1	1			
Multiple atypical resections	126	83	23	20	
Left	13/126 (10.3%)	6		7	
Right	36/126 (28.6%)	22		14	
Bilateral	77/126 (61.1%)	54	23		
Right bi-segmentectomy or hemihepatectomy with left atypical	5	1	4		

**Table 2 cancers-15-00435-t002:** Per-operative outcomes for patients who underwent multiple laparoscopic liver resections.

Article	Tumor Maximum Size (mm)	*p* (vs. Single)	Mean Blood Loss (mL Range)	*p* (vs. Single)	Mean Operative Time (min (Range)	*p* (vs. Single)	Conversion Rate	*p* (vs. Single)
Karazyan et al. [20] (n = 104)	22	0.12	300 (50–5000)	0.75	186 (75–390)	0.26	2.9%	0.41
D’Hondt et al. [21] (n = 36)	Not described		250 (150–450)	<0.001, higher for multiple	200 (170–230)	<0.001, longer for multiple	8.3%	0.07
Nassar et al. [22] (n = 39)	23.9	0.69	188.9 (0–1000)	0.39	217.3 (90–369)	0.039, longer for multiple	0%	0.88

**Table 3 cancers-15-00435-t003:** Description of available laparoscopic guidance techniques for CRLM resections.

Technique	Location of Tumors Detected	Tumor Margins	Detection of Missing Tumors from Preoperative CT Scan	Availability in OR	Disadvantages
Intraoperative ultrasound	Superficial or deep	Location relative to veins	Yes	Available	Technicality
Indocyanine green fluorescence	Only superficial	Real-time visualization	Yes, if superficial	Available	No deep lesion visualization
3D models	Superficial and deep	Location relative to anatomical structures	No	Not available	Location of tumor detected only if by preoperative CT-scan

**Table 4 cancers-15-00435-t004:** Postoperative short- and long-term outcomes for patients who underwent multiple laparoscopic liver resection.

Article	90-Days Morbidity Rate	*p* (vs. Single)	Major Complication (Clavien III–IV)	*p* (vs. Single)	90-Days Mortality Rate (n)	*p* (vs. Single)	Length of Stay (Days)	*p* (vs. Single)	5-Year OS	*p* (vs. Single)	5-Year RFS	*p* (vs. Single)
Karazyan et al. [20] (n = 104)	20 (19.2%)	0.17	14 (13.4%)	Not described	1 (0.96%)		3 (1–26)	0.62	42%	0.62	16%	0.14
D’Hondt et al. [21] (n = 36)	2 (5.6%)	1.0	1 (2.8%)	1.0	0 (0%)	1.0	5 (4–7)	0.015, longer for multiple	66%	0.49	28%	0.62
Nassar et al. [22] (n = 39)	14 (27.5%)	0.82	1 (1.9%)	0.45	0 (0%)	0.94	5.5	0.59	Not described		Not described	

**Table 5 cancers-15-00435-t005:** Data for laparoscopic two-stage hepatectomy.

Article	1st Stage Laparoscopy	2nd Stage Laparoscopy	Interval Time (Months)	Mean Number of CRLM	Conversion Rate 1st/2nd	90-Day Morbidity Rate 1st/2nd
Fuks et al. [42]	34	26	3		2/4	
Okumura et al. [43]	38	38	2.8	6	1/4	16%/26%
Kilburn et al. [44]	7	1	3.4	5.2	0/1	0%/0%
Taillieu et al. [45]	23	7	1.9	Not described	1/1	0%/14%
Di Fabio et al. [46]	8	3	2.9	4	0/1	0%/Not described
Görgec et al. [47]	Not described	12	Not described	3.6	-/2	Not described/17%
Levi Sandri et al. [48]	5	0	2.2	6.6	0/-	Not described

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
