# Peer review of "Multiple Laparoscopic Liver Resection for Colorectal Liver Metastases"

_cancers, 2023, doi:10.3390/cancers15020435_

Round 1
Reviewer 1 Report
Multiple laparoscopic liver resection is one of the challenging surgery, and I enjoyed the authors' manuscript.
However, the description is insufficient as a review.
1. There are few TABLES, the summary is difficult to understand.
2. In FEASIBILITY OF MULTIPLE CONCOMITANT LLR, the aurhors cited the papers reported by Russolillo et al(19) and J Kalil et al(14), but did not say anything about the main points of these papers.
3. The authors summarized and analyzed cases from only three reports(20-22). A systemic review and meta-analysis should be done.
4. IOUS is sometimes difficult in LLR. The accuracy of IOUS in LLR compared to open surgery, especially in multiple CRLM cases, needs more discussion. The authors should discuss about that.
5. It should be mentioned that it is impossible to identify deep lesions even with the use of ICG.
6. In "POSTOPERATIVE SHORT-TERM OUTCOMES AFTER MULTIPLE LLR" and "ONCOLOGICAL OUTCOMES AFTER MULTIPLE LLR FOR CRLM", the readers would rather know the comparison between laparoscopic and open multiple liver resections than that between laparoscopic single resection and multiple resections, because it is important to determine which we should apply, LLR or open liver resection for multiple CRLM. Additionally, the authors should summarize the review results using Tables.
7. In POSTOPERATIVE SHORT-TERM OUTCOMES AFTER MULTIPLE LLR, the authors described that "multiple LLR seems to lower the risk of postoperative liver failure. However, that depends on the remnant liver volume and liver function. This description is unfounded.
8. The authors described that "Compared to open TSH(39,41), laparoscopic TSH had significantly less blood less, significantly shorter length of hospital stay,・・・". Were those results about 2nd hepatectomy?. Or both of 1st and 2nd?
9. In multiple liver resections, I'm wondering how many tumors can be safely removed laparoscopically. When the feasilbility of multiple resections is dicussed, the authors should discuss that.
Author Response
Manuscript ID: cancers-2087593
Title: Multiple laparoscopic liver resection for colorectal liver metastases
Reviewer #1
Please see the attachment

Reviewer 2 Report
This review is important to summarize the state of the art of hepatic resection in colorectal liver metastases. However, the exposition is confusing and not always very clear. My recommendation is to revise the wording of the text and the English language to make the work more understandable
Author Response
Manuscript ID: cancers-2087593
Title: Multiple laparoscopic liver resection for colorectal liver metastases
Reviewer #2
Please see the attachment

Round 2
Reviewer 1 Report
The authors described multiple LLR for CRLM, and I enjoyed reading this manuscript. They revised the manuscript, and It has improved well.
The points below should be well addressed.
1. In the section 3, "This training requires routine use of IOUS..." is not necessary.
2. In the section 4, the listing of Table 3 is incorrect for Table 4.
3. Since multiple LLR usually takes longer than open surgery, especially in many resection cases, there is a limitation on the number of resections in laparoscopic surgery. The authors should discuss the indication for LLR with regard to the maximum number of resections.
4. In the section 2, how many cases with CRLM was included in the study from Russolillo and et al(19)?  How many is the highest number of resections in the study?
5.The authors integrate the four studies(20-23) in table 1. The cases registered in these studies probably have different patient backgrounds and indications for LLR, and it would not make sense to integrate them. They could be summarized separately.
6. In the section 2, "In the right lobe, 3 patients underwent multiple anterior LLR..." is unclear. Describe more clearly.
7.In table2 and table 4, the authors should describe the number of cases and the results of comparison with single resection in each study.
8. In the section 2 and section 6, "In our experience, ..."
They should include specific data, including the number of cases, or cite their existing reports.
9. In the section 7, I do not think the description "The number of tumors or resections do not seem to be an obstacle to perform the liver resection by laparoscopy" is correct. They should clarify the indications for LLR in the existing reports with regard to the number of tumors or resections.
Author Response
Thank you for your comments. Please see the attachment for the point-by-point reply.

Author Response

(The authors gave the same response as above.)
